# 5-HT Receptors and the Development of New Antidepressants

**DOI:** 10.3390/ijms22169015

**Published:** 2021-08-20

**Authors:** Grzegorz Ślifirski, Marek Król, Jadwiga Turło

**Affiliations:** Department of Drug Technology and Pharmaceutical Biotechnology, Faculty of Pharmacy, Medical University of Warsaw, 1 Banacha Street, 02-097 Warsaw, Poland; gslifirski@wum.edu.pl (G.Ś.); jturlo@wum.edu.pl (J.T.)

**Keywords:** antidepressants, drug design, dual 5-HT_1A_/SERT activity, multimodal activity

## Abstract

Serotonin modulates several physiological and cognitive pathways throughout the human body that affect emotions, memory, sleep, and thermal regulation. The complex nature of the serotonergic system and interactions with other neurochemical systems indicate that the development of depression may be mediated by various pathomechanisms, the common denominator of which is undoubtedly the disturbed transmission in central 5-HT synapses. Therefore, the deliberate pharmacological modulation of serotonergic transmission in the brain seems to be one of the most appropriate strategies for the search for new antidepressants. As discussed in this review, the serotonergic system offers great potential for the development of new antidepressant therapies based on the combination of SERT inhibition with different pharmacological activity towards the 5-HT system. The aim of this article is to summarize the search for new antidepressants in recent years, focusing primarily on the possibility of benefiting from interactions with various 5-HT receptors in the pharmacotherapy of depression.

## 1. Introduction

Depression is a mental illness that affects over 250 million people worldwide [1]. Emotional (depressed mood, irritability, anhedonia), somatic (sleep, appetite, libido), and functional disorders (suicidal thoughts, slowed speech and movement, learning, memory and attention deficits) [2] make this disease the main cause of disabilities in the general population [3,4].

An important step in the treatment of depressive disorders is the introduction of SSRIs (serotonin reuptake inhibitors), which are currently first-line antidepressants (e.g., fluoxetine, sertraline, escitalopram). Their mechanism of action is based on the serotonergic system, and the molecular target is the serotonin transporter protein (SERT). The effectiveness of these therapeutics, unfortunately, leaves much to be desired; 60–70% of patients do not experience a remission of symptoms, and 30–40% do not respond to the treatment at all [5]. A serious drawback of selective serotonin reuptake inhibitors is their latency period, i.e., a delay in the therapeutic response by 2–6 weeks. Common side effects for SSRIs are sexual dysfunction, anxiety, and food intolerances.

Apart from SSRIs, other selective monoamine reuptake inhibitors are also used in pharmacotherapy. Reboxetine, a selective norepinephrine reuptake inhibitor, appears to be less effective than the SSRIs. These observations may, however, result from its relatively low tolerance [6]. Bupropion, on the other hand, is a norepinephrine and dopamine reuptake inhibitor and, therefore, has a more activating profile than SSRI drugs. Two drugs, venlafaxine and duloxetine, are classified as dual serotonin-norepinephrine reuptake inhibitors (SNRIs). However, the efficacy of the norepinephrine reuptake blocking at clinical doses of duloxetine is unclear [7]. Clinical guidelines often recommend the use of SNRIs in patients who do not respond to SSRIs [8,9,10].

There is a need for the further exploration of the neurochemical causes of depression. Recent studies report the influence of many various types of neurosignaling on the mechanism of depression [11,12,13,14]. The search for new generations of antidepressants using the triple reuptake inhibition mechanism (SSRI/SNRI/SDARI), or the combination of serotonin reuptake inhibition with affinities for various 5-hydroxytryptamine (5-HT) receptor subtypes, broadens the knowledge in this field [15,16,17].

A significant part of recent studies proves that serotonergic dysfunction, especially related to the postsynaptic 5-HT_1A_ receptor, plays an important role in the pathomechanism of Major Depressive Disorder (MDD) [18,19,20,21,22,23,24]. Clinical trials show that the combination of SSRIs with both partial agonism and antagonism of the 5-HT_1A_ receptor may result in an improvement in the speed and efficacy of the antidepressant effect [23,25,26]. This can be confirmed by the drugs recently introduced into the pharmacotherapy of depression–vilazodone and vortioxetine (Figure 1). Vilazodone exhibits partial agonist activity at the 5-HT_1A_ receptor, while vortioxetine binds to several 5-HT receptor subtypes (5-HT_1A_, 5-HT_1B_, 5-HT_1D_, 5-HT_3_, and 5-HT_7_). For example, the degree of sexual dysfunction associated with the use of vilazodone has been found to be relatively low [27]. Vortioxetine, on the other hand, positively influences cognitive impairment related to depression [10,28].

The targeted pharmacological modulation of serotonergic transmission in the brain continues to be a leading strategy in the search for new antidepressants. The careful selection of molecular targets for the proper use of the mechanisms of serotonergic modulation, which influences other neurotransmission systems, seems to be the most effective strategy for supplementing the activity of “serotonin-enhancing” drugs in the near future. A better understanding of the receptors and receptor signaling responsible for the effects of serotonin on neurogenesis can also help in the development of new and more effective drugs. The aim of this article is to summarize the search for new antidepressants in recent years, focusing primarily on the possibility of benefiting from interactions with various 5-HT receptors in the pharmacotherapy of depression.

## 2. The Serotonergic System and Depression

Serotonin, or 5-hydroxytryptamine (5-HT), is a monoamine neurotransmitter found throughout the human body [19,29]. Serotonin is synthesized in the midbrain in a small population of raphe nucleus neurons where tryptophan hydroxylase is expressed [30]. However, serotonin synthesis is not limited to the central nervous system (CNS), as tryptophan hydroxylase is also found in enterochromaffin cells in the gastrointestinal tract [31]. In fact, it should be noted that most of the serotonin in the human body is produced by this cell type [32]. Serotonin binds to more than 14 receptor proteins, most of which are G-protein coupled receptors [30,33]. This molecule mediates the transmission of several physiological and cognitive systems throughout the body that affect emotions, memory, sleep, and thermal regulation [34].

Serotonin is synthesized in the body from an essential amino acid—L-tryptophan. Ingested with food, L-tryptophan is converted into serotonin through a series of reactions. The first step, which simultaneously limits the rate of serotonin synthesis, is the hydroxylation of L-tryptophan to 5-hydroxy-L-tryptophan (5-HTP) by tryptophan hydroxylase (TPH) using oxygen and tetrahydropteridine as co-factors. There are two isoforms of TPH that can participate in this reaction: TPH1, expressed predominantly peripherally; and TPH2, expressed only in the brain. L-aromatic amino acid decarboxylase (AADC) then converts 5-HTP to serotonin [19,31].

The crossing of the blood–brain barrier (BBB) by serotonin is impossible due to its acid dissociation [35]; therefore, the amount of serotonin present in the CNS depends on the amount of centrally present L-tryptophan. The L-tryptophan present in the systemic circulation is actively transported by the BBB to the CNS using a carrier protein, where it is then converted into serotonin. Serotonin synthesized in the central nervous system is stored in secretory vesicles, where it remains until neuronal depolarization triggers its release into the synaptic cleft, allowing postsynaptic binding. Once released into the synapse, the serotonin molecules are eventually taken up by the serotonin transporter (5-HTT), which is located on the presynaptic axonal membrane. After the above-mentioned reuptake occurs, serotonin molecules are metabolized by monoamine oxidase (MAO) to 5-hydroxyindole acetic acid (5-HIAA) [29]. There are two isoforms of MAO (MAO-A and MAO-B), and both break down serotonin into neurons through oxidative deamination. The serotonin metabolite (5-HIAA) is actively transported from the CNS to the periphery and then excreted in the urine [19].

Already by the 1950s, it was noted that several mental illnesses showed abnormalities in the serotonergic system. The relationship between the serotonergic system and depression has been confirmed in clinical trials. They showed that an acute, transient relapse of depressive symptoms can be produced in subjects in remission using p-chlorophenylalanine (an irreversible inhibitor of serotonin synthesis). L-tryptophan depletion, causing a temporary reduction in central serotonin levels, had similar consequences. These findings have shown that the clinical efficacy of antidepressants depends on the presynaptic serotonergic function. Other studies have demonstrated a reduced concentration of the major metabolite of serotonin (5-HIAA) in the cerebrospinal fluid of untreated depressed patients and a reduced concentration of 5-HT and its major metabolite (5-HIAA) in the postmortem brain tissue of depressed and/or suicidal patients [20].

The serotonergic neurons of the mammalian brain constitute the most extensive and complex neurochemical network in the CNS after the glutamatergic system, which is the brain’s primary transmission network. It has been estimated that the human brain contains approximately 250,000 5-HT neurons. For comparison, the total number of all neurons is around 10^11^ [36]. While serotonergic neurons originate mainly in the brainstem dorsal and median raphe nuclei, they arborise over large areas such that they innervate almost every area of the brain with high densities of axonal varicosities. Some serotonergic projections create classical chemical synapses, but many release 5-HT in a paracrine manner (sometimes referred to as “volumetric transmission”). In addition, serotonin neurons exhibit slow (~1 Hz) and regular tonic activity that ceases during the rapid eye movement sleep phase (REM-off neurons). This activity is parallel to the noradrenergic neurons of the locus coeruleus [34]. Under normal conditions, the activity of serotonergic neurons is tightly controlled by a number of mechanisms, including: (i.) glutamatergic inputs from the forebrain (mainly the prefrontal cortex) [37], (ii.) the tonic noradrenergic input from the pontine nuclei [38], (iii.) inhibitory GABAergic signals from local interneurons [39], and (iv.) dopamine signals from the dopaminergic nuclei of the midbrain [40]. Moreover, the serotonin system is, in a way, self-regulating. The key control mechanism of 5-HT neurons is negative feedback through the 5-HT_1A_ autoreceptors [20]. This mechanism is currently being studied in great detail in the context of the treatment of CNS diseases.

The aforementioned anatomical and electrophysiological picture shows that changes in the activity of serotonergic neurons affect a large population of target neurons in the forebrain. The complex nature of the serotonergic system and interactions with other neurochemical systems indicate that the development of MDD may be mediated by various pathomechanisms. Currently suggested mechanisms include: (i.) low neuronal production of serotonin or of postsynaptic receptors, (ii.) decreased excitatory inputs or excessive system self-control, and (iii.) decreased 5-HT synthesis and/or tryptophan deficiency. The common denominator of these phenomena in depression is undoubtedly the disturbed transmission in the central 5-HT synapses. Therefore, the deliberate pharmacological modulation of serotonergic transmission in the brain seems to be one of the appropriate strategies for the search for new antidepressants.

## 3. The 5-HT Receptors

The serotonergic system affects various physiological functions, including psychoemotional expression, sensorimotor integration, and the regulation of the autonomic, cardiovascular, respiratory, and digestive systems. Within the CNS, 5-HT is involved in the regulation of higher mental functions and emotions, extrapyramidal motor functions, and cognitive functions (e.g., learning and memory).

At least 14 different serotonin receptors have been identified. These receptors can be divided into distinct families, which are labelled 1, 2, 3, 4, 5, 6, and 7, and the subtypes in each family are labelled with letters (e.g., a, b, c). Many of these receptors are thought to be involved in the pathogenesis of various CNS disorders [41].

### 3.1. The 5-HT_1A_ Receptors

The 5-HT_1A_ receptors are located primarily in the following populations: (i.) presynaptic neurons of the raphe nuclei of the midbrain and (ii.) postsynaptic neurons, mainly in the hippocampus, septum, amygdala, and corticolimbic regions [42]. Autoreceptors are located within the bodies and dendrites of serotonin neurons. Their activation inhibits neuronal discharges and reduces the release of serotonin [43]. Thus, 5-HT_1A_ autoreceptors play an important role in the self-regulation of the serotonergic system; they partially inhibit the activity of adenylate cyclase [44] and activate G protein-dependent rectifying potassium channels (GIRK) with the use of the βγ subunit of G protein [45]. This causes membrane hyperpolarization, a reduction in neuronal excitability, and the inhibition of potential-dependent calcium channels, reducing the influx of calcium ions. The consequence is a reduction in the neural discharge rate. Given the significant influence of these neuronal discharges on the overall activity of the entire serotonergic system, it can be concluded that the reduction in the firing rate evoked by serotonin and other 5-HT_1A_ agonists immediately translates into an overall reduction in 5-HT release in most areas of the brain, particularly in regions innervated by the dorsal raphe [20].

The activation of 5-HT_1A_ autoreceptors by endogenous serotonin, therefore, plays an essential role in the physiological control of the activity of the 5-HT ascending neurons. The 5-HT neurons during waking periods show a slow and regular rate of discharge [36]. Under conditions of excessive excitatory input (e.g., stress), there is an increased release of serotonin in the vicinity of neuronal bodies. It activates 5-HT_1A_ autoreceptors, which allow low and regular neuronal activity to be maintained [40]. Thus, 5-HT_1A_ autoreceptors act as negative feedback physiological “safety valves” to maintain homeostasis.

The expression of 5-HT_1A_ heteroreceptors, in turn, takes place in populations of non-serotonin receptors, mainly in the limbic system within: (i.) bodies and dendrites of glutamatergic neurons [43] or (ii.) axons of GABA-ergic [46], and (iii.) cholinergic neurons [47]. These receptors are involved in regulating the release of various neurotransmitters: acetylcholine in the medial septum [48], glutamate in the prefrontal cortex [49], and dopamine in the ventral tegmental area [50]. In most regions of the brain, the inhibition of adenylate cyclase occurs due to the activation of the Gαi protein. The GIRK channels in the hippocampus are activated by the βγ subunits of the Gαo isoform [51]. The 5-HT_1A_ receptors in the cortex and hypothalamus bind to both the Gαi and Gαo subunits, while their preferential binding to the Gαi3 protein occurs within the raphe nucleus.

The differences in the properties of 5-HT_1A_ auto- and hetero-receptors are manifested in their different functional selectivity [52]: 5-HT_1A_ heteroreceptors stimulate [53], while 5-HT_1A_ autoreceptors inhibit ERK1/2 transmission [54]. The 5-HT_1A_-biased agonism appears to result in the preferential activation of a specific signaling pathway without affecting or even blocking other pathways associated with this receptor subtype [55]. It has also been shown that there is an agonist-dependent modulation of G-protein coupling and a transduction of 5-HT_1A_ receptors in rat dorsal raphe nucleus. Moreover, 8-hydroxy-2-(di-n-propylamino)tetralin (8-OH-DPAT, a full 5-HT_1A_ receptor agonist) compared with buspirone (a partial 5-HT_1A_ receptor agonist) fails to modify forskolin-stimulated cAMP accumulation [56].

In general, 5-HT_1A_ receptor-deficient mice show a shorter immobility time in the forced swim test than wild-type control animals [57]. The lack of functional 5-HT_1A_ autoreceptors may, therefore, favor a less-depressed phenotype. The whole-life suppression of 5-HT_1A_ heteroreceptor expression in adolescence results in a broad depression-like phenotype. In addition, the group showed physiological and cellular changes within medial prefrontal cortex–dorsal raphe proper circuitry: (i.) increased basal serotonin levels in the medial prefrontal cortex, which is hyporeactive to stress and (ii.) decreased basal serotonin levels and firing rates in a dorsal raphe hyperactivated by the same stressor [57].

Animal studies show that both the stimulation and blockade of 5-HT_1A_ receptors can cause or accelerate the antidepressant effect [17]. It is difficult not to associate this with the above-described functional differences of 5-HT_1A_ auto- and hetero-receptors and the phenomenon of the biased 5-HT_1A_ agonism. Many studies have demonstrated the antidepressant effect of 8-OH-DPAT reversed by 5-HT_1A_ receptor antagonists [58]. Moreover, 5-HT_1A_ receptor-deficient mice showed no increase in adult neurogenesis in the hippocampus after chronic treatment with fluoxetine (SSRI) and not with imipramine (TCA) [59]. The preferential activation of postsynaptic 5-HT_1A_ receptors by F15599 (Figure 2), a biased 5-HT_1A_ agonist, resulted in an antidepressant-like effect [60]. Similar activity was shown by F13714, a non-selective agonist of 5-HT_1A_ receptors, but it induced a deeper “serotonin syndrome”, hypothermia, and corticosterone release in rats. Elevated corticosterone levels accompany chronic stress in animals, leading to depression [61]. Moreover, the activation of 5-HT_1A_ receptors in the prefrontal cortex (PFC) by F15599 produces strong antidepressant-like effects in the forced swim test (FST) in rats, with a distinctive bimodal dose–response pattern. These data suggest that F15599 may target specific 5-HT_1A_ receptor subpopulations in the PFC, possibly located on the GABAergic and/or glutaminergic neurons [62].

The previously described physiological function of 5-HT_1A_ autoreceptors and their regulation of depressive behavior seem to be unfavorable in the context of the mechanism of action of antidepressants [20,63]. The negative feedback pathway through 5-HT_1A_ autoreceptors may decrease the efficacy of the SSRI as the dose increases, thus creating a second, anomalous part of the dose–response curve. This effect may also be responsible for the so-called therapeutic window for such antidepressants [64]. The prolonged use of SSRIs translates into significantly higher levels of extracellular 5-HT than after a single administration [65]. The negative feedback loop is believed to be the cause of the slow and delayed clinical efficacy of antidepressant drugs [66]. Administration of antidepressants (tricyclic drugs, monoamine oxidase inhibitors, and SSRIs) significantly increases the level of extracellular 5-HT in the midbrain raphe [67]. This leads to: (i.) the activation of 5-HT_1A_ receptors, (ii.) the reduction in 5-HT cell firing [68], and (iii.) the terminal release of 5-HT [69]. The inhibition of SSRIs in the negative feedback pathway clearly decreases with the duration of treatment. This is most likely due to the serotonin-induced desensitization of raphe 5-HT_1A_ autoreceptors discussed earlier [70]. Thus, the desensitization of 5-HT_1A_ autoreceptors may accelerate the onset and/or enhance the antidepressant effect [71]. Mice with higher levels of 5-HT_1A_ autoreceptors showed a blunted physiological response to acute stress, increased behavioral despair, and no behavioral response to fluoxetine [72]. Moreover, mice with lower autoreceptor levels showed a strong behavioral response to fluoxetine after both chronic and subchronic administration [72]. Thus, lowering the level of 5-HT_1A_ autoreceptors prior to antidepressant treatment may accelerate and increase the effectiveness of antidepressant therapy. Combining SSRI treatment with the 5-HT_1A_ receptor antagonist pindolol significantly reduces the latency of the antidepressant response and improves the clinical response in previously untreated MDD patients (Table 1) [20,21,73]. The above data indicate that the stimulation of postsynaptic 5-HT_1A_ receptors or the blockade of presynaptic 5-HT_1A_ receptors results in antidepressant-like activity. (-)-pindolol may also stimulate somatodendritic 5-HT_1A_ receptors. Then, its accelerating antidepressant effect might stem from the accelerated adaptive changes like autoreceptor desensitization in response to both serotonin and pindolol. This mechanism can also be achieved by initiating the treatment with high-dose SSRI when a patient is suicidal. The antidepressant action of pindolol may also be related to its agonistic activity at the β_1_-adrenoreceptor as this drug possesses the strongest intrinsic sympathicomimetic activity among other β-blockers [74].

According to the neurotrophic hypothesis of depression, decreased neurotrophic support causes neuronal atrophy, which in turn reduces hippocampal neurogenesis and leads to depression. Clinical data support this theory: postmortem analysis has shown reduced volumes of the hippocampus and prefrontal cortex in depressed patients [78,79]. Persons diagnosed with MDD showed decreased levels of BDNF (brain-derived neurotrophic factor) and NGF (nerve growth factor) in the hippocampus. A deficit of these neurotrophins may promote neuronal loss [80,81]. This phenomenon was confirmed by in vivo studies [82,83,84,85], which showed that antidepressants reversed these changes [86]. Chronic treatment with 8-OH-DPAT, in turn reduced the feeding delay in the novelty-suppressed feeding test and increased adult hippocampal neurogenesis in wild-type mice, but showed no effect in the 5-HT_1A_ receptor knockout group [59]. Thus, 5-HT_1A_ receptors mediate the action of 8-OH-DPAT, from which it can be concluded that the postsynaptic 5-HT_1A_ receptors mediate the antidepressant-like action of 8-OH-DPAT [87]. The specific deletion of the 5-HT_1A_ heteroreceptors from mature granular cells in the dentate gyrus of the hippocampus has also been found to abolish the effects of SSRIs in various behavioral tests [88]. It also attenuated the effects of SSRIs on adult neurogenesis and the expression of hippocampal neurotrophic factors: BDNF and VEGF (vascular endothelial growth factor). Whole-life 5-HT_1A_ heteroreceptor-knockout (but not autoreceptor-knockout) mice showed decreased mobility in the forced swim test [89]. Such a depression-like phenotype was not observed when the suppression of heteroreceptors was initiated in adulthood. Therefore, serotonergic signaling in the forebrain during development may stably influence the circuits underlying the behavioral response to the FST [89].

The STAR*D clinical trial shows that in patients unsuccessfully treated with SSRIs, the augmentation with buspirone resulted in symptom remission [75]. Buspirone (a partial agonist of the 5-HT_1A_ receptor) enhances the desensitization of 5-HT_1A_ autoreceptors, increasing the effectiveness of the SSRI treatment. Recently, a single transcription factor, Freud-1, has been found to be crucial for the expression of the 5-HT_1A_ autoreceptor [90]. Mice with a conditional knockout of Freud-1 in serotonin neurons were shown to have elevated levels of 5-HT_1A_ autoreceptors and exhibited the enhanced anxiety and depressive behavior in adulthood that was refractory to chronic SSRI treatment [90]. Interestingly, the double knockout of the Freud-1/5-HT_1A_ gene did not produce such effects. In this case, the depressive-like behavior was even reduced [90]. The study suggests that targeting specific transcription factors may increase the response to antidepressant treatment. These reports indicated the need to search for compounds targeting only the population of 5-HT_1A_ auto- or heteroreceptors.

The results of postmortem and neuroimaging studies suggest an increased density of 5-HT_1A_ autoreceptors in patients with MDD compared to the control group [91,92,93]. Genetic studies have shown that individuals with an increased density or activity of 5-HT_1A_ autoreceptors are more prone to mood disorders and respond poorly to antidepressant treatment [94,95]. However, the number and density of postsynaptic 5-HT_1A_ receptors have been shown to be unaltered or reduced in depressed patients, and this alteration is not sensitive to antidepressant treatment [96]. Long-term antidepressant therapy causes the tonic activation of 5-HT_1A_ receptors in the dorsal hippocampus [97], and activation of 5-HT_1A_ receptors in the dentate gyrus increases hippocampal neurogenesis [98]. In light of the cited reports, the use of 5-HT_1A_ agonists as antidepressants seems natural [99]. Some agents possessing such activity (e.g., buspirone and gepirone) show antidepressant efficacy in placebo-controlled trials, but their potency is lower than that of SSRIs. Most 5-HT_1A_ agonists (especially azapirones, Figure 3) show the preferential activation of presynaptic 5-HT_1A_ receptors. Moreover, these agents tend to have a reduced efficacy at postsynaptic 5-HT_1A_ receptors. Thus, endogenous serotonin competes in the postsynaptic sites with an exogenous substance (with lower agonism), which causes a paradoxical reduction in the tone at the postsynaptic 5-HT_1A_ receptors. Higher doses of 5-HT_1A_ agonists (such as those used in experimental animals) are likely to result in the greater activation of postsynaptic 5-HT_1A_ receptors, which may explain the positive results of efficacy studies in animal models. Conversely, the administration of the selective 5-HT_1A_ receptor antagonist DU125530 with fluoxetine did not accelerate or increase the efficacy of fluoxetine in a double-blind, randomized, placebo-controlled clinical trial. DU125530 had similar binding to pre- and post-synaptic 5-HT_1A_ receptors [100], and the blockade of postsynaptic 5-HT_1A_ receptors likely offset the benefits of enhancing presynaptic serotonergic function [101]. This may show the importance of the activation of postsynaptic 5-HT_1A_ receptors in the mechanism of antidepressant action.

Observations on the 5-HT_1A_ receptor population contributed to a fruitful search for potential multimodal antidepressants that incorporate 5-HT_1A_ receptor activity into their mechanism of action [102]. Recently developed compounds seem to overcome the aforementioned therapeutic problems of azapirones and other first-generation 5-HT_1A_ agonists. Two new antidepressants, vilazodone [27,103] and vortioxetine [104,105], inhibit 5-HT reuptake and show the partial agonism at 5-HT_1A_ receptors.

The 5-HT_1A_ receptor ligands also possess their own potentially therapeutic activity. The 5-HT_1A_ partial agonists show antianxiety [106,107], antidepressant [108], antiaggressive [109], anticraving [110], and anticataleptic properties [111]:Animal studies show that both the stimulation and blockade of 5-HT_1A_ receptors can cause or accelerate the antidepressant effect. It is difficult not to associate this with the functional differences of 5-HT_1A_ auto- and hetero-receptors and the phenomenon of a biased 5-HT_1A_ agonism;A single transcription factor, Freud-1, has been found to be crucial for the expression of the 5-HT_1A_ autoreceptor. Targeting it may increase the response to antidepressant treatment;Observations on the 5-HT_1A_ receptor population contributed to a fruitful search for potential multimodal antidepressants (vilazodone and vortioxetine) that incorporate 5-HT_1A_ receptor activity into their mechanism of action.

### 3.2. The 5-HT_1B_ Receptors

The 5-HT_1B_ receptors, like 5-HT_1A_ receptors, are located pre- and post-synaptically and are also negatively coupled to adenylate cyclase. Their highest densities are in the striatum, pallidum, nucleus accumbens, substantia nigra, and ventral tegmental area. Lower levels of 5-HT_1B_ receptors are found in the hippocampus, amygdala, and cingulate cortex [112].

Unlike somatodendritic 5-HT_1A_ autoreceptors, 5-HT_1B_ autoreceptors are located on serotonergic axons, where they regulate the synthesis and release of 5-HT locally. The 5-HT_1B_ postsynaptic receptors are located mainly in the centers of motor control (such as the basal ganglia), where they control the synaptic transmission of other neurotransmitters [112]. Studies have shown that 5-HT_1B_ receptors play a role in depression, anxiety, migraines, locomotor activity, aggressive behavior, and the potentiation of the action of other drugs [112,113,114].

Animal studies show that the involvement of 5-HT_1B_ receptors in the pathophysiology of depression is partly related to their responsiveness to environmental stress as well as their exposure to antidepressants [115]. The 5-HT_1B_ heteroreceptors are involved in hippocampal neurogenesis, which may explain their importance for the antidepressant-like effect [116]. Mice lacking 5-HT_1B_ autoreceptors showed an increased mobility in the FST as well as an increased preference for lower-sucrose concentrations in the sucrose preference test compared to the control group. After SSRI administration, elevated levels of serotonin in the hippocampus were observed [117]. Moreover, two common genetic polymorphisms of 5-HT_1B_ receptors, G861C [118] and C129T [119], were associated with MDD and affective disorders. The 5-HT_1B_ receptor gene knockout mice showed increased aggression [120].

The p11 protein, which colocalizes with 5-HT_1B_ and 5-HT_4_ receptors [121], plays a key role in modulating the function of the 5-HT_1B_ receptor. Its dysregulation has been reported in preclinical models of depression and in postmortem samples from MDD patients [122]. The p11 protein improves 5-HT_1B_ receptor function in various regions of the brain and contributes to an antidepressant-like effect in animal behavioral tests [123]. P11 knockout mice showed depression-like behavior and demonstrated a reduced responsiveness to 5-HT_1B_ receptor agonists and tricyclic antidepressants [123].

Studies in the learned helplessness model showed that 5-HT_1B_ receptors were upregulated in various regions of the brain following stress exposure. A reduced 5-HT_1B_ autoreceptor function and, thus, increased serotonin release, has also been demonstrated after chronic antidepressant treatment [124]. Moreover, chronic treatment with SSRIs induced a negative regulation and/or desensitization of 5-HT_1B_ autoreceptors [125] and facilitated the effect of SSRIs in serotonin neurotransmission [126]. Compounds exhibiting 5-HT_1B_ antagonism, administered alone or with antidepressants, have been shown to be effective in preclinical models of depression [127]. The pretreatment with 5-HT_1B_ receptor antagonists [128] or the genetic inactivation of the 5-HT_1B_ receptor [129] increased the SSRI-induced effect in mice. Therefore, the blockade of 5-HT_1B_ autoreceptors may promote the antidepressant effect. It has been suggested that the 5-HT_1B_ receptor antagonists themselves may be attributed to an antidepressant-like effect. SB-616234-A, a 5-HT_1B_ receptor antagonist, decreased immobility in a forced swim test in mice (Figure 4) [130]. The selective 5-HT_1B_ receptor inverse agonist, SB236057A, increased, in turn, the extracellular concentration of serotonin in the dentate gyrus of a guinea pig. This effect was comparable to that of 14 days of paroxetine therapy [131]. The acute blockade of the 5-HT_1B_ receptor might cause a rapid antidepressant effect [131]. It appears that the agonist activation of 5-HT_1B_ heteroreceptors may also induce antidepressant-like effects [132]. CP94253, a selective 5-HT_1B_ receptor agonist, showed an antidepressant-like activity in a forced swimming test in mice [133]. Anpirtoline, as a selective 5-HT_1B_ receptor agonist, also reduced immobility in control mice but had no effect in 5-HT_1B_ knockout mice [132]. The effect of this compound in the FST was, therefore, due to the activation of the 5-HT_1B_ receptor. The above studies suggest that 5-HT_1B_ receptors play a role in antidepressant-like activity. Ther stimulation of postsynaptic receptors and the inhibition of presynaptic 5-HT_1B_ receptors may be beneficial in the treatment of depression [134].

As with 5-HT_1A_ receptors, acute SSRI therapy activates terminally localized 5-HT_1B_ receptors, thus reducing 5-HT synthesis and release. The long-term administration of SSRIs desensitizes terminal 5-HT_1B_ autoreceptors [135], suggesting that the plasticity of the autoregulatory function of both 5-HT_1A_ and 5-HT_1B_ receptors may be important with respect to the therapeutic profile of SSRIs. Again, as with 5-HT_1A_ receptor antagonists, the administration of 5-HT_1B_ receptor antagonists increases the neurochemical and behavioral effects of SSRIs [128,136]. Interestingly, the co-administration of the selective 5-HT_1A_ antagonist WAY-100635 and the 5-HT_1B_ receptor antagonist SB-224289 has an additive effect, enhancing the neurochemical effects of fluoxetine. This has led to the suggestion that the combination of the 5-HT_1A_ and 5-HT_1B_ receptor antagonism may increase CNS serotonin levels and, therefore, potentially be an effective treatment strategy for depression [20]:Animal studies show that the involvement of 5-HT_1B_ receptors in the pathophysiology of depression is partly related to their responsiveness to environmental stress as well as an exposure to antidepressants;The p11 protein improves 5-HT_1B_ receptor function in various regions of the brain and contributes to an antidepressant-like effect in animal behavioral tests;The 5-HT_1B_ heteroreceptors are involved in hippocampal neurogenesis, which may explain their importance for the antidepressant-like effect. The stimulation of postsynaptic receptors and the inhibition of presynaptic 5-HT_1B_ receptors may be beneficial in the treatment of depression.

### 3.3. The 5-HT_1D_, 5-HT_1E_, and 5-HT_1F_ Receptors

The clinical significance of the remaining 5-HT_1_ receptors (5-HT_1D_, 5-HT_1E_, 5-HT_1F_) is less clear. There is limited preclinical evidence linking some of the receptors with depressive states. The sensitivity of postsynaptic 5-HT_1D_ receptors in patients after treatment with SSRIs has been found to be impaired [137]. On the other hand, a postmortem study of untreated suicidal victims with a confirmed history of depression showed a much higher density of 5-HT_1D_ receptors in the globus pallidus [138]. The observed high expression of the 5-HT_1E_ receptor in the frontal cortex and hippocampus may indicate the relationship between 5-HT_1E_ receptors and cognitive functions and memory [20,139].

### 3.4. The 5-HT_2A_ Receptors

The 5-HT_2A_ receptors, like the others of the 5-HT_2_ family, are preferentially coupled to the G protein of the Gq/11 type, so their activation increases the cellular level of inositol phosphate and, consequently, the cytosolic concentration of calcium ions. The 5-HT_2A_ receptors are distributed postsynaptically and presynaptically throughout the brain at serotonergic terminals, with the greatest concentration in the neocortex [140,141,142]. Recent anatomical and functional studies suggest that 5-HT_2A_ receptors are also present presynaptically as heteroreceptors, where they may enhance glutamatergic neurotransmission and participate in memory processes [143]. It has also been demonstrated that the 5-HT_2A_ receptors of the cerebral cortex are located on GABAergic interneurons as well as glutamatergic projection neurons in the brains of humans and rodents [42,144].

Many antidepressants and antipsychotic drugs possess a relatively high binding to 5-HT_2A_ receptors [145]. Although there is no direct correlation between the affinity of these drugs for 5-HT_2A_ receptors and clinically effective doses, there is ample evidence that the 5-HT_2A_ receptor plays a role in the pathomechanism of depression [20,146]. Some antidepressants mediate their action partly via the antagonism of 5-HT_2A_ receptors [147]. In addition, chronic treatment with antidepressants, such as tricyclic antidepressants, monoamine oxidase inhibitors, mianserin, mirtazapine, or sertraline, decreased the number of 5-HT_2A_ receptors in rodents [148]. Chronic electroconvulsive shock treatment resulted in the upregulation of cortical 5-HT_2A_ receptors in rodents [149].

Several clinical trials have shown that atypical antipsychotics [150] and the antidepressant mirtazapine with an affinity for α_2_-adrenoceptors and 5-HT_2A_ receptors [151] augment the clinical response to SSRIs in treatment-resistant patients [76]. A common feature of these substances is their ability, at clinical doses, to block responses to signals mediated by 5-HT_2A_ receptors [152]. Such downregulation could, inter alia, explain why the side effects of SSRIs diminish after 2 or 3 weeks. The high co-expression of 5-HT_1A_ and 5-HT_2A_ receptors in the neocortex [153] may indicate that the blockade of 5-HT_2A_ receptors enhances 5-HT_1A_ receptor-mediated neurotransmission in the cortical and limbic regions, an activity associated with antidepressant efficacy. The chronic administration of 5-HT_2A_ receptor antagonists has been shown to result in a paradoxically negative regulation of 5-HT_2A_ receptors [154,155], which may be beneficial in the treatment of depression. Moreover, preclinical studies indicate that 5-HT_2A_ antagonists have anxiolytic properties, as demonstrated by ritanserin, a 5-HT_2A_ antagonist with anxiolytic effects in humans [156].

Another issue is the relationship between the 5-HT_2A_ receptor and the noradrenergic system in relation to depression [157]. Studies have shown that the activation of 5-HT_2A_ receptors as a result of treatment with SSRIs causes an increase in serotonin levels in GABA neurons. This inhibits the neuronal activity of norepinephrine through the prolonged release of GABA [158,159,160]. In turn, citalopram, in addition to reducing norepinephrine firing, also has the effect of lowering basal and evoked extracellular norepinephrine levels in the amygdala [161]. This may underlie SSRI ineffectiveness in resistant depression. The co-administration of an SSRI and a 5-HT_2A_ receptor antagonist trazodone (as well as atypical antipsychotics, such as quetiapine, risperidone, olanzapine, and aripiprazole) reversed this inhibitory effect in noradrenergic neurons in rats and might be beneficial in the treatment of resistant depression [160,162,163,164]. Increasing evidence shows that 5-HT_2A_ receptor antagonists display antidepressant effects. EMD 281014 (Figure 5), a 5-HT_2A_ receptor antagonist, showed significant activity in the FST in congenital learned helpless rats [165]. A similar effect was shown by another 5-HT_2A_ receptor antagonist, FG5893, which significantly shortened the immobility time in the FST [166]. The selective 5-HT_2A_ receptor antagonist, M100907, enhanced the antidepressant-like behavioral effects of fluoxetine [167], suggesting that a selective 5-HT_2A_ receptor blockade may complement the behavioral effects of serotonin transporter inhibition. In contrast, recent studies in rats have shown that the functional disturbance of the 5-HT_2A_ receptor in the medial prefrontal cortex may contribute to postpartum mental disorders, including depression and psychosis [168]. In addition, prefrontal 5-HT_2A_ receptors may both have beneficial and negative effects on cognition, which might explain the aggravation of cognitive deficits after the onset of SSRI treatment in depressed patients, as well as the limited efficacy of second-generation antipsychotics that act as 5-HT_2A_ receptor antagonists against the strongly debilitating cognitive symptoms of schizophrenia and other psychiatric disorders [169]. A deficiency in 5-HT_2A_ receptors has also been shown to alter the metabolic and transcriptional, but not behavioral, consequences of chronic unpredictable stress in mice [170]. The 5-HT_2A_ blockade or SSRI-induced downregulation of 5-HT_2A_ may lead to emotional blunting in patients. It is, therefore, very likely that 5-HT_2A_ receptors may have different functions depending on the region of the brain:

Many antidepressants and antipsychotic drugs have relatively high binding to 5-HT_2A_ receptors;The high co-expression of 5-HT_1A_ and 5-HT_2A_ receptors in the neocortex may indicate that the blockade of 5-HT_2A_ receptors enhances 5-HT_1A_ receptor-mediated neurotransmission in the cortical and limbic regions, an activity associated with antidepressant efficacy;Increasing evidence shows that 5-HT_2A_ receptor antagonists display antidepressant effects. A selective 5-HT_2A_ receptor blockade may complement the behavioral effects of serotonin transporter inhibition.

### 3.5. The 5-HT_2B_ Receptors

The 5-HT_2B_ receptor is expressed mainly in peripheral tissues, especially in the liver, kidneys, and heart, and its distribution in the brain is low [171]. In the central nervous system, the 5-HT_2B_ receptor is present in septal nuclei, the dorsal hypothalamus, and the medial amygdala at levels similar to those found in the stomach [171]. The 5-HT_2B_ receptor, mRNA, is found in the dorsal raphe nucleus, suggesting a potential autoreceptor role [172]. The 5-HT_2B_ receptors are coupled to the Gq protein, which activates PLC (phospholipase C)/PKC (protein kinase C) and increases the concentration of calcium ions in the cytosol.

The knowledge about the function of the 5-HT_2B_ receptor in the CNS is limited; however, there are reports of the antidepressant properties of selective 5-HT_2B_ receptor agonists [173]. The presence of 5-HT_2B_ receptors in the dorsal raphe and their stimulatory role in 5-HT release has been demonstrated [173]. The pharmacological or genetic inactivation of the 5-HT_2B_ receptor abolished the effects of chronic treatment with SSRIs, and the stimulation of 5-HT_2B_ receptors induced an SSRI-like response in behavioral and neurogenic tests. In turn, the genetic inactivation of 5-HT_2B_ receptors in serotonergic neurons eliminated the neurogenic effects of fluoxetine [173]. It has recently been confirmed that 5-HT_2B_ receptors directly and positively regulated the activity of serotonin neurons [174]. In addition, the stimulation of the 5-HT_2B_ receptor via fluoxetine in astrocyte cell cultures resulted in the phosphorylation of extracellular signal-regulated kinases and the transactivation of the EGF (epidermal growth factor) receptor [175]. A reduced level of astroglial (but not neuronal) 5-HT_2B_ receptors in a mouse model of Parkinson’s disease was also reported, which paralleled the development of the depression-like phenotype [176]. The stimulation of astroglial 5-HT_2B_ receptors may, therefore, be beneficial in treating depressive disorders [177].

Considering the role of peripherally located 5-HT_2B_ receptors, potential new antidepressants acting on 5-HT_2B_ receptors may adversely affect the function of the respiratory and circulatory systems [17,20]:5-HT_2B_ receptors directly and positively regulate the activity of serotonin neurons;There are reports of the antidepressant properties of selective 5-HT_2B_ receptor agonists;Potential new antidepressants acting on 5-HT_2B_ receptors may adversely affect the function of the respiratory and circulatory systems.

### 3.6. The 5-HT_2C_ Receptors

The 5-HT_2C_ receptors are mainly located in the choroid plexuses, cerebral cortex, hippocampus, substantia nigra, and cerebellum. They bind preferentially with Gq/11 and increase the concentrations of inositol phosphates and cytosolic Ca^2+^. Like 5-HT_2A_ receptors, they are involved in the regulation of mood, motor behavior, and appetite [178].

Several classes of antidepressants have an affinity for 5-HT_2C_ receptors. Although these receptors are usually somatodendritic, in some regions they are also present on axon terminals [179]. The location of 5-HT_2C_ receptors in relation to serotonergic and GABAergic neurons in the anterior raphe nuclei demonstrates complex systemic relationships in the brain. It has been shown that 5-HT_2C_ receptors are preferentially located on GABAergic interneurons (and not on serotonergic neurons). This suggests that the stimulation of GABAergic interneurons by 5-HT_2C_ receptors plays an important role in the suppression of serotonergic cell firing in the dorsal raphe and surrounding areas [180]. The immunoreactivity of the 5-HT_2C_ receptor has also been described in GABAergic cells in the PFC [181] and in the dopaminergic and GABAergic neurons of the mesolimbic pathway [182].

A potent 5-HT_2C_ receptor antagonist, S32006 (Figure 6), showed antidepressant activity in rodent behavioral tests and increased dopamine and norepinephrine levels in the frontal cortex [183]. This compound reduced immobility in the FST in mice, suppressed anhedonia in a chronic mild stress model, and increased cell proliferation and BDNF expression in the dentate gyri of rats [183]. In contrast, the inverse agonist of the 5-HT_2C_ receptor, S32212, showed an antidepressant effect in the FST in rats after both acute and chronic treatment [184].

On the other hand, some studies report that 5-HT_2C_ agonists have been shown to be active in animal models of depression, suggesting an antidepressant-like effect [185,186]. WAY-163909, a selective 5-HT_2C_ receptor agonist, elicited a rapid antidepressant effect in a rat FST that was blocked by the 5-HT_2C_/_2B_ receptor antagonist, SB206553 [186]. Moreover, after chronic treatment, WAY163909 reduced the hyperactivity associated with olfactory bulbectomy in rats [186,187]. It is possible that the mediated antidepressant effects of these compounds were due to the stimulation of 5-HT_2C_ receptors and the resulting activation of postsynaptic serotonin receptors [188]. Other selective 5-HT_2C_ receptor agonists have also been effective in animal models of depression and obsessive–compulsive disorder [189].

Preclinical data show that the antagonism of 5-HT_2C_ receptors increases the neurochemical and behavioral effects of SSRIs. Examples include: the increase in the effect of SSRIs on extracellular 5-HT concentrations in the hippocampus and cortex [190,191], or a significant increase in the effect of SSRIs in behavioral models of depression by selective and non-selective 5-HT_2C_ antagonists [190].

Additionally, 5-HT_2C_ receptors have been shown to be involved in the anti-immobility effect of antidepressants in the FST, increasing the serotonin level in the synapse [192]. Few studies suggest that 5-HT_2C_ receptor antagonists alone may also exhibit antidepressant-like properties. The inactivation of 5-HT_2C_ receptors has been shown to potentiate SSRI-induced serotonin release in rodents [190]. However, 5-HT_2C_ receptor antagonists administered separately had no effect on serotonin levels [191].

An altered editing of the mRNA-encoding 5-HT_2C_ receptors has been reported in the PFC of depressed suicide victims [193]. The desensitization of these receptors has been observed in patients after chronic treatment with SSRIs [194].

The 5-HT_2C_ receptors are also involved in the tonic modulation of dopaminergic activity [195]. The role of the dopaminergic system in schizophrenia, along with the antagonism of atypical antipsychotics towards the 5-HT_2C_ receptors, has aroused interest in this receptor for the treatment of schizophrenia [196]. Conversely, the ineffectiveness of SSRIs in some patients may be due to the serotonin-related inhibition of the neuronal activity of dopamine in the ventral capping region via 5-HT_2C_ receptors [157]. Escitalopram has been shown to reduce the stimulation of dopamine neurons by activating 5-HT_2C_ receptors located on GABA neurons. Some studies indicate that the co-administration of SSRIs with 5-HT_2C_ receptor antagonists (including atypical antipsychotics, such as aripiprazole) may eliminate the inhibitory effects on dopaminergic neurons in rat brains and restore the effect of the SSRI [163]. The aforementioned 5-HT_2C_ receptor antagonist S32006, with a potential antidepressant- and anxiolytic-like effect, increased dopamine levels in the frontal cortex of rats and enhanced dopaminergic neuron firing [183]. The modulation of dopaminergic activity may, therefore, be beneficial in the development of antidepressants due to the above-mentioned activity of 5-HT_2C_ receptor ligands. Recent studies demonstrate the contradictory effect of 5-HT_2C_ receptors on the effects of SSRIs on motor function and affective behavior, highlighting the potential benefits of 5-HT_2C_ receptor antagonists both for reducing SSRI motor side effects and enhancing the therapeutic antidepressant and anxiolytic effects [197].

Both 5-HT_2C_ receptor agonists and antagonists exhibit antidepressant-like activity, and there is still a need to further define the role of this receptor subtype in depression:Several classes of antidepressants have an affinity for 5-HT_2C_ receptors. Alterations in their functional status have been observed in depressive and anxiety states;Both 5-HT_2C_ agonists and antagonists have been shown to be active in animal models of depression. Preclinical data show that the antagonism of 5-HT_2C_ receptors increases the neurochemical and behavioral effects of SSRIs;There is still a need to further define the role of 5-HT_2C_ receptor subtype in depression.

### 3.7. The 5-HT_3_ Receptors

The activation of the 5-HT_3_ receptor leads to a rapid opening of the transmembrane channel, resulting in an increase in the conductivity of Na^+^/K^+^ ions and an immediate influx of extracellular Ca^2+^ ions. This, in turn, triggers the release of neurotransmitters and/or peptides. The 5-HT_3_ receptors are found throughout the brain and CNS and the highest density of 5-HT_3_ receptors was found in the spinal cord and brainstem. The 5-HT_3_ receptors present in the dorsal vagal complex are involved in the control of the emetic mechanism [142]. Many 5-HT_3_ receptor antagonists have been developed as antiemetics for use in cancer chemotherapy. In the 1990s, lithoxetine, an antidepressant combining serotonin reuptake and 5-HT_3_ receptor antagonism, was developed to prevent SSRI-induced gastrointestinal side effects [198]. In the forebrain, on the other hand, 5-HT_3_ receptors were present mainly in structures of the limbic system, such as the hippocampus, amygdala, and entorhinal cortex [199].

The 5-HT_3_ receptors are involved in the control of dopamine and acetylcholine release. They also control the functioning of the GABAergic system. Activity towards other neurotransmission systems is the main mechanism of action for 5-HT_3_ receptor ligands. The 5-HT_3_ receptors are expressed on different types of GABAergic interneurons in the forebrain [200,201]. The physiological stimulation of serotonergic neurons stimulates cortical (and possibly hippocampal) GABAergic neurons. This likely results in the inhibition of neighboring excitatory neurons by GABA_A_ and GABA_B_ receptors [201].

Preclinical studies suggest that the 5-HT_3_ receptor plays a role in mental disorders [17]. The 5-HT_3_ antagonists show antidepressant-like activity in various animal models [202]. The systemic administration of tropisetron (a 5-HT_3_ receptor antagonist) prevented restraint stress-induced dopamine release in the nucleus accumbens and prefrontal cortex in rats. This suggested that 5-HT_3_ receptors mediated the stress-dependent activation of dopaminergic neurotransmission [203]. Tropisetron additionally exerted an antidepressant-like effect in FST in rats. This effect was abolished after a pretreatment with mCPGB (1-(m-chlorophenyl)-biguanide), a potent 5-HT_3_ receptor agonist (Figure 7) [204].

Some antidepressants with different mechanisms of action exhibit functional 5-HT_3_ receptor antagonism [205]. Chronic treatment with fluoxetine desensitizes 5-HT_3_ receptors [206], and SERT knockout mice show increased 5-HT_3_ receptor density compared to wild-type mice [207]. It has been suggested that the antidepressant effect of SSRIs is partially dependent on the blockade of 5-HT_3_ receptors [208]. The relatively new multimodal antidepressant drug vortioxetine [209,210] displays nanomolar binding affinities to the SERT (K_i_ = 1.6 nM) and other serotonin receptors, including 5-HT_3_, 5-HT_1A_, 5-HT_7_, 5-HT_1B_ and 5-HT_1D_, with K_i_ values of 3.7 nM, 15 nM, 19 nM, 33 nM and 54 nM, respectively [102]. Vortioxetine antagonism at the 5-HT_3_ receptor [211] may underlie its faster onset of action [212]. Rodent experiments show that the antidepressant-like effect should be attributed to postsynaptic, rather than presynaptic, 5-HT_3_ antagonism, since the presynaptic and somatodendritic 5-HT_3_ receptor blockade reduces serotonin levels [203]. The antidepressant and/or anxiolytic effects recently demonstrated by some 5-HT_3_ receptor antagonists in animal models of depression may result from the modulation of the hypothalamic–pituitary-adrenal axis, interaction with the serotonergic system, or antioxidant properties [213,214,215,216,217,218].

The agonism of the 5-HT_3_ receptor reduces the antidepressant effect in the FST in rats [219], while the antagonism of the 5-HT_3_ receptor reduces the immobility time in the FST [220]. Ondansetron, a 5-HT_3_ receptor antagonist, confirms these observations; it exhibits antidepressant properties in the TST (tail suspension test) and FST, also enhancing the effect of fluoxetine [220]. In a model of chronic unpredictable stress in mice, the administration of ondansetron reversed depressive behavior affecting the hypothalamic-pituitary-adrenal axis [221]. Moreover, in mice with streptozotocin-induced diabetes, the drug displayed antidepressant and anxiolytic properties, possibly through the antagonism of the 5-HT_3_ receptor [222]. Behavioral studies with ondansetron (and tropisetron) also suggested an interaction of 5-HT_3_ and NMDA receptors, as well as an involvement of the nitric oxide-cyclic guanosine monophosphate pathway inhibition in the observed antidepressant-like effects [223,224]. Studies on genetically modified animals confirm the role of 5-HT_3_ receptors in the antidepressant effect. The 5-HT_3_ receptor knockout mice were reported to display an antidepressant-like phenotype [225].

In vitro electrophysiology studies showed that low-dose citalopram treatment desensitized the 5-HT_1A_ receptor only in the dorsal raphe nucleus of 5-HT_3_ knockout mice, while high dose treatment caused similar 5-HT_1A_ autoreceptor desensitization in 5-HT_3_ knockout and wild types [225]. Hence, lower doses of citalopram may be effective when 5-HT_3_ receptors are deactivated. It has also been shown that the blockade of the 5-HT_3_ receptor by ondansetron enhances the effect of citalopram on extracellular serotonin levels in the rat forebrain [102]. The use of combined SSRIs and 5HT_3_ receptor antagonists is proposed as an improvement strategy to be tested in the treatment of depressive disorders [226]. The neurochemical, electrophysiological, and behavioral consequences of the repeated administration of this drug combination will need to be assessed.

The properties of 5-HT_3_ receptor antagonists have also been used to alleviate substance abuse, which is often associated with most psychiatric disorders, including MDD [227]. The 5-HT_3_ receptor antagonists have been reported to be effective in reducing ethanol and morphine intake [228]. It is worth noting that various antipsychotics are non-competitive 5-HT_3_ receptor antagonists, and this may contribute to their efficacy [229]. There is likely to be an association between 5-HT_3_ receptors and anxiety behavior [230]. The 5-HT_3_ antagonists reverse helpless behavior in rats [231] and abolish the emotion-potentiated startle effect in humans [232]:Activity towards other neurotransmission systems is the main mechanism of action for 5-HT_3_ receptor ligands;Rodent experiments show that the antidepressant-like effect should be attributed to postsynaptic rather than presynaptic 5-HT_3_ antagonism, since the presynaptic and somatodendritic 5-HT_3_ receptor blockade reduces serotonin levels;Some antidepressants with different mechanisms of action exhibit functional 5-HT_3_ receptor antagonism. The vortioxetine antagonism at the 5-HT_3_ receptor may underlie its faster onset of action.

### 3.8. The 5-HT_4_ Receptors

The 5-HT_4_ receptors in the CNS are mainly located in the putamen, caudate nucleus, hippocampus, nucleus accumbens, globus pallidus, and substantia nigra. To a lesser extent, these receptors are present in the neocortex, raphe and pontine nuclei, and thalamus [233]. Studies using positron emission tomography show a slightly more limited regional distribution of 5-HT_4_ receptors in the human brain, showing a high density of this receptor in the caudate–putamen and much lower densities in the frontal cortex and hippocampus [234]. The 5-HT_4_ receptors are coupled to the Gs protein, which activates adenylate cyclase/PKC and increases the intracellular level of cAMP. Regarding peripheral tissues, these receptors play an important role in the heart, gastrointestinal tract, adrenal glands, and urinary bladder [235].

There are reports linking the 5-HT_4_ receptor with depressive disorders [236]. Preclinical models of depression, such as the olfactory bulbectomized and glucocorticoid heterozygous receptor mice, show that the expression of 5-HT_4_ receptors increased in the ventral hippocampus or striatum, respectively [237], while in the Flinders-sensitive line rat model of depression, the downregulation of 5-HT_4_ receptors was observed in the ventral and dorsal hippocampus [238].

The 5-HT_4_ receptor subtype is involved in the modulation of synaptic plasticity [239], which is influenced by antidepressants [240]. The signaling of the 5-HT_4_ receptor may modulate the function of the dentate gyrus of the hippocampus by increasing the neurogenesis and expression of neurotrophic factors, which may contribute to the antidepressant effects of drugs that enhance serotonergic transmission [241]. The 5-HT_4_ receptor interacts with the p11 protein, which determines the antidepressant activity mediated by 5-HT_1B_ and 5-HT_4_ receptors [242].

In addition, 5-HT_4_ knockout mice show an enhanced response of serotonergic neurons to citalopram [243]. Thus, 5-HT_4_ receptors are possibly involved in the activation of 5-HT neurons during SERT inhibition. As observed for 5-HT_1A_ and 5-HT_2A_ receptors [37,40], 5-HT_4_ receptors in the PFC control the firing rate of midbrain serotonergic neurons via descending inputs [244]. In addition, 5-HT_4_ receptors mediate synaptic transmission between the dentate gyrus and the CA3 field of the hippocampus. Fluoxetine was observed to normalize the mossy fiber pathway by activating 5-HT_4_ receptors [245]. Chronic treatment with fluoxetine and venlafaxine (but not reboxetine) decreased the 5-HT_4_ receptor density in rat brain [246]. Although the 5-HT_4_ receptor antagonist, SB 204070A, showed no independent effect and did not reduce the immobility time in the FST in naive rats [247], another receptor antagonist, GR 125487, blocked fluoxetine activity in a mouse corticosterone-induced depression model [248]. Therefore, this study suggests that the activation of 5-HT_4_ receptors mediates the antidepressant-like effects of fluoxetine. It has been shown that a knockout of the 5-HT_4_ receptor can induce some adaptive changes in mice, leading to depression and anxiety-like behavior. Moreover, 5-HT_4_ receptor knockout mice do not respond to fluoxetine in the olfactory bulbectomized model of depression and anxiety [249]. On the other hand, some studies suggest that the behavioral effects of fluoxetine in the corticosterone-induced model of depression and anxiety do not appear to be dependent on 5-HT_4_ receptors [250].

Preclinical studies show that the administration of the 5-HT_4_ agonists, RS67333 and prucalopride (Figure 8), reduces the immobility time in the FST, thus demonstrating the potential of the 5-HT_4_ receptor as a molecular target of a potential new generation of antidepressants [251]. The agonism of the 5-HT_4_ receptor may also play a role in the cognitive deficits associated with MDD. The use of RS67333 in chronic neuroendocrine animal models of depression/anxiety resulted in the restoration of induced learning and memory disorders [252]. Moreover, the studies show that administration of RS67333 and prucalopride causes 5-HT_1A_ autoreceptor desensitization, increased the tonus on hippocampal postsynaptic 5-HT_1A_ receptors, and increased CREB phosphorylation and neurogenesis in the hippocampus [251]. These parameters, which characterize the functioning of the brain, are used in antidepressant therapies. Importantly, these effects are noticeable after 3 days of treatment [251], while they are usually only seen after 2–3 weeks of treatment with SSRIs due to the latency phenomenon. The faster response to 5-HT_4_ agonism has been suggested to be a result of the parallel rapid and sustained activation of 5-HT neuronal firing in the dorsal raphe nucleus [253]. Increased serotonergic neuronal firing may also underlie the apparently superior efficacy of 5-HT_4_ agonists over SSRIs because the reuptake inhibitory effect depends on the basal rates of 5-HT cell firing. Since the 5-HT_4_ receptor is not expressed in the raphe nuclei, the ability of 5-HT_4_ receptors to stimulate the firing of 5-HT neurons appears to involve the activation of receptors located on neurons in the PFC [244]. The identity of the cells expressing 5-HT_4_ receptors and their connections to the serotonergic neurons of the dorsal raphe nucleus are not yet well understood. It is possible that they project to other regions, contributing to the antidepressant effect of 5-HT_4_ agonists [253].

The activation of the 5-HT_4_ receptor may be a useful adjunct to antidepressant therapy, both to accelerate the onset of clinical antidepressant effects and to target cognitive symptoms that are not effectively treated with current therapies [254]:There are reports linking the 5-HT_4_ receptor with depressive disorders. The 5-HT_4_ receptor interacts with the p11 protein, which determines the antidepressant activity mediated by 5-HT_1B_ and 5-HT_4_ receptors;5-HT_4_ receptor signaling may modulate the function of the dentate gyrus of the hippocampus by increasing the neurogenesis and expression of neurotrophic factors, which may contribute to the antidepressant effects of drugs that enhance serotonergic transmission.The activation of the 5-HT_4_ receptor may be a useful adjunct to antidepressant therapy, both to accelerate the onset of clinical antidepressant effects and to target cognitive symptoms that are not effectively treated with current therapies.

### 3.9. The 5-HT_6_ Receptors

The 5-HT_6_ receptors are the postsynaptic receptors most expressed in the striatum, nucleus accumbens, olfactory tubercle, and cortex. They are also moderately dense in the amygdala, hippocampus, hypothalamus, thalamus, and cerebellum [255].

This serotonin receptor subtype has been found to play a role in learning and memory [256] as well as in the central regulation of hunger and satiety behavior [257]. The 5-HT_6_ receptors may, therefore, serve as a novel molecular target for the improvement of cognitive functions [258]. Several of the tricyclic antidepressants (e.g., amitriptyline) and atypical antidepressants (e.g., mianserin) exhibit nanomolar 5-HT_6_ binding and antagonistic activity [259]. This fact, as well as the distribution of the 5-HT_6_ receptor in the limbic and cortical regions of the brain, may suggest that 5-HT_6_ receptors play an important role in the pathogenesis and/or treatment of depression [259]. The 5-HT_6_ antagonists (SB-399885, Figure 9) show antidepressant activity in the FST and in the TST in rodents (rats and mice) [260]. In addition, the combination of an ineffective dose of SB-399885 with ineffective doses of imipramine, desipramine, bupropion, or moclobemide has been shown to exert antidepressant effects in the rat FST [261]. This suggests that the inhibition of the 5-HT_6_ receptor potentiates the effects of clinically used antidepressants. This synergistic effect is interesting in the search for a multimodal antidepressant therapy with minimized side effects or a faster onset of action. Other preclinical studies show that 5-HT_6_ agonism can be used in the treatment of depression. The reduction in the immobility of mice in the FST after the administration of WAY208466, a selective potent agonist of the 5-HT_6_ receptor, has been demonstrated [262]. The 5-HT_6_ receptor partial agonist, EMD386088, caused antidepressant- and anxiolytic-like effects after intrahippocampal administration [263]. This also occurred after acute and chronic treatment in rats [264], possibly because it directly stimulated the receptor. The stimulation of the 5-HT_6_ receptor may initiate the biochemical and behavioral effects induced by SSRIs (fluoxetine) [122]. On the other hand, the 5-HT_6_ receptor agonist LY-586713 increases the expression of BDNF (a marker of cellular antidepressant activity) in the hippocampus after just a single administration [265]. In comparison, SSRIs require multiple applications to produce the same effect [240]. Therefore, it is unclear what functional 5-HT_6_ receptor ligand profile (antagonism or agonism) will be more beneficial in the treatment of depression. Moreover, the exact mechanism by which 5-HT_6_ ligands induce antidepressant effects is unknown and may include effects on other neurotransmission systems [260,266]:

The distribution of the 5-HT_6_ receptor in the limbic and cortical regions of the brain may suggest that 5-HT_6_ receptors play an important role in the pathogenesis and/or treatment of depression;Several tricyclic and atypical antidepressants exhibit nanomolar 5-HT_6_ receptor binding. The inhibition of the 5-HT_6_ receptor potentiates the effects of clinically used antidepressants. This synergistic effect is interesting in the search for a multimodal antidepressant therapy with minimized side effects or a faster onset of action;It is not exactly clear what functional 5-HT_6_ receptor ligand profile (antagonism or agonism) will be more beneficial in the treatment of depression. Moreover, the exact mechanism by which 5-HT_6_ ligands induce antidepressant effects is unknown and may include effects on other neurotransmission systems.

### 3.10. The 5-HT_7_ Receptors

The 5-HT_7_ receptor is highly expressed in the thalamus, hypothalamus, hippocampus, and cortex [267]. The results of immunolocation and autoradiography studies are generally consistent with the pattern of mRNA distribution [268,269], suggesting a dominant somatodendritic localization. The physiological role of 5-HT_7_ receptors is to regulate circadian rhythm, sleep, and mood [270].

As with the 5-HT_6_ receptors, several antidepressants [271] and antipsychotics [272] have been found to have a high affinity for the 5-HT_7_ receptor, leading to much further research into its antidepressant activity. One preclinical study in rats showed that several antidepressants, both tricyclic and SSRIs, induce *c-fos* expression in a manner consistent with 5-HT_7_ receptor activation within the suprachiasmatic nucleus, and that chronic treatment with antidepressant drugs downregulates 5-HT_7_ receptor binding [271].

Preclinical studies also indicate the antidepressant and anxiolytic effects of the selective 5-HT_7_ receptor antagonist, SB-269970, in rodents [273], as well as a synergistic interaction between subeffective doses of this agent and antidepressants, leading to a reduction in immobility in both the FST and the TST [274,275]. The intrahippocampal administration of SB-269970 (Figure 10) induced an antidepressant effect in the FST in rats [276]. The co-administration of citalopram and SB-269970 increased the activity of serotonin neurons in rats and improved the antidepressant effect in the TST [274]. SB-269970 enhanced the antidepressant effect of antidepressants (citalopram, imipramine, desipramine, and moclobemide) in the FST in mice [275]. It was also shown that the administration of SB-269970 for only one week caused the behavioral, electrophysiological, and neuroanatomical changes that usually occur after a long-term treatment with SSRIs. Therefore, 5-HT_7_ receptor antagonists might represent a new class of antidepressants with a faster therapeutic effect. JNJ-18038683, another 5-HT_7_ receptor antagonist, was also effective in mice TST [277]. Moreover, the compound potentiated serotonin transmission, REM suppression, and antidepressant-like behaviour induced by citalopram in rodents [277]. The above studies indicate that the participation of the 5-HT_7_ receptor in the antidepressant-like action, and blockade of the 5-HT_7_ receptor may not only induce but accelerate this action.

In addition, the genetic and pharmacological inactivation of 5-HT_7_ receptors partially reversed phencyclidine-induced deficits of pre-pulse inhibition, an animal model for antipsychotic activity [278]. Similarly, it should also be noted that the atypical antipsychotic aripiprazole, which has a high affinity for the 5-HT_7_ receptor, is sometimes used to enhance the effects of traditional antidepressants [279]. Similarly, there are reports that the antidepressant effect of amisulpride is mediated by its action on 5-HT_7_ receptors [280].

Vortioxetine is a high affinity inhibitor of the human 5-HT transporter, 5-HT_3_ and 5-HT_7_ receptors, and a 5-HT_1A_ agonist [281]. Although the affinity of vortioxetine for the rat 5-HT_7_ receptor is lower compared to the human receptor [282], subacute administration (within 3 days) of an effective dose of vortioxetine rapidly lowers rat 5-HT_7_ receptor levels [283]. This preclinical evidence suggests that vortioxetine has a relatively low affinity for the 5-HT_7_ receptor compared to other 5-HT receptor subtypes but inhibits its action with a rapid 5-HT_7_ receptor downregulation as an inverse agonist, similar to other 5-HT_7_ receptor-inhibiting mood-stabilizing atypical antipsychotics: clozapine, lurasidone, and olanzapine [283,284]. In other words, the rapid-acting antidepressant and anxiolytic actions of 5-HT_7_ receptor antagonism are worth reassessing in the context of drug development after future clinical data have been accumulated. Overall, the 5-HT_7_ receptor is currently considered a promising target for the development of antidepressants [285]. Recent clinical studies have shown that both the intravenous and oral administration of vortioxetine resulted in a significant improvement in depression (Montgomery Åsberg Depression Rating Scale and Hospital Depression Scale) and anxiety (Hospital Anxiety Scale) after 3 days [77]:Several antidepressants have been found to have a high affinity for the 5-HT_7_ receptor, leading to much further research into its antidepressant activity;The antagonists of the 5-HT_7_ receptor might represent a new class of antidepressants with a faster therapeutic effect.Preclinical evidence suggests that vortioxetine has a relatively low affinity for the 5-HT_7_ receptor compared to other 5-HT receptor subtypes but inhibits its action with a rapid 5-HT_7_ receptor downregulation as an inverse agonist. Both the intravenous and oral administration of vortioxetine resulted in a significant improvement in depression and anxiety after 3 days.

## 4. Conclusions

The development of new antidepressants is based on monoamine systems. The targeted pharmacological modulation of serotonergic transmission in the brain continues to be a leading strategy in the search for new antidepressants. As can be seen from this review, the serotonergic system offers great potential for the development of new antidepressant therapies based on the combination of SERT inhibition with different pharmacological activities towards the 5-HT system. The careful selection of molecular targets for the proper use of the mechanisms of serotonergic autoregulation and selective/biased activation or the blockade of relevant receptors (e.g., stimulation of postsynaptic 5-HT_1A_, postsynaptic 5-HT_1B_, 5-HT_2B_ and 5-HT_4_ receptors; or the blockade of presynaptic 5-HT_1A_, presynaptic 5-HT_1B_, 5-HT_2A_, 5-HT_3_, and 5-HT_7_), which also influences other neurotransmission systems, seems to be the most effective strategy for supplementing the activity of “serotonin-enhancing” drugs in the near future. A better understanding of receptors and the receptor signaling responsible for the effects of serotonin on neurogenesis could also help in the development of new and more effective drugs.

## Figures and Tables

**Figure 1 ijms-22-09015-f001:**
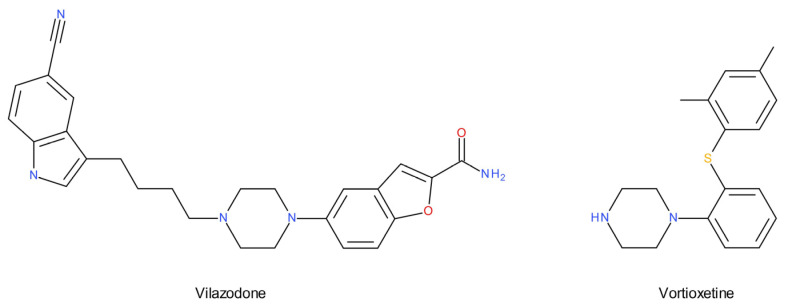
Novel antidepressants: vilazodone and vortioxetine.

**Figure 2 ijms-22-09015-f002:**
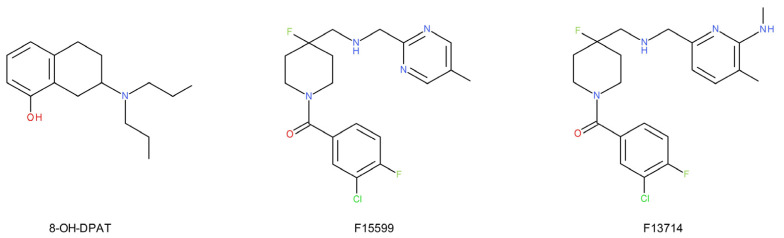
5-HT_1A_ receptor agonists: 8-OH-DPAT, F15599, and F13714.

**Figure 3 ijms-22-09015-f003:**
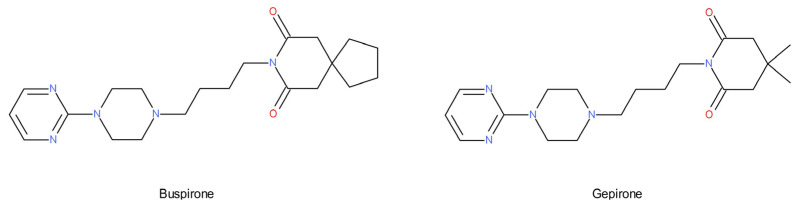
Azapirones: buspirone and gepirone.

**Figure 4 ijms-22-09015-f004:**
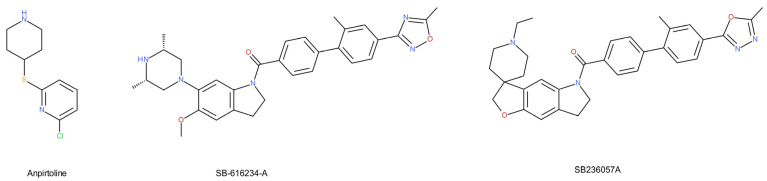
5-HT_1B_ receptor ligands: anpirtoline, SB-616234-A, and SB236057A.

**Figure 5 ijms-22-09015-f005:**
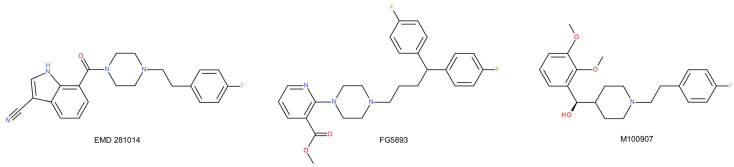
5-HT_2A_ receptor antagonists: EMD 281014, FG5893 and M100907.

**Figure 6 ijms-22-09015-f006:**
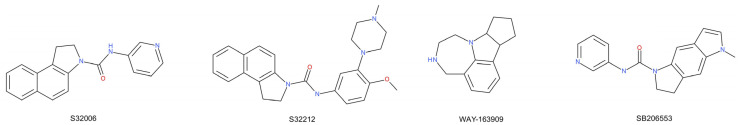
5-HT_2C_ receptor ligands: S32006, S32212, WAY-163909 and SB206553.

**Figure 7 ijms-22-09015-f007:**
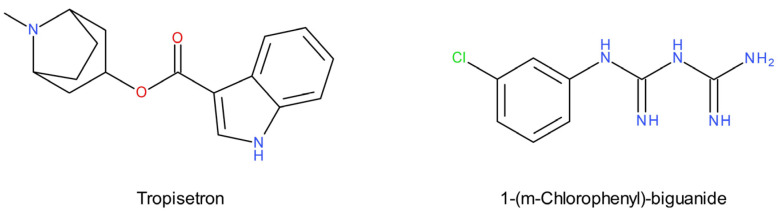
5-HT_3_ receptor ligands: tropisetron and mCPGB.

**Figure 8 ijms-22-09015-f008:**
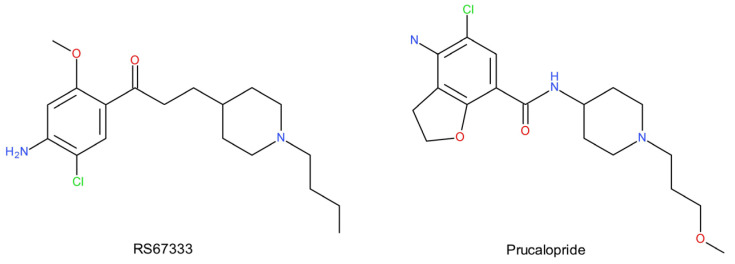
5-HT_4_ receptor agonists: RS67333 and prucalopride.

**Figure 9 ijms-22-09015-f009:**
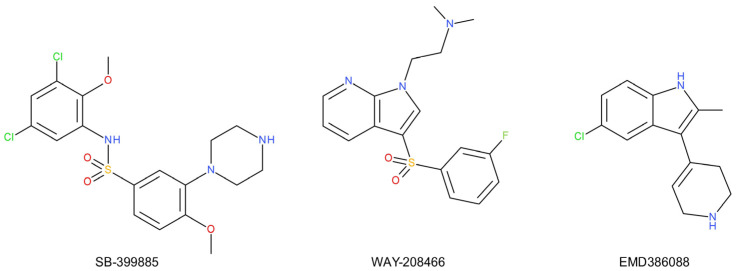
5-HT_6_ receptor ligands: SB-399885, WAY-208466, and EMD386088.

**Figure 10 ijms-22-09015-f010:**
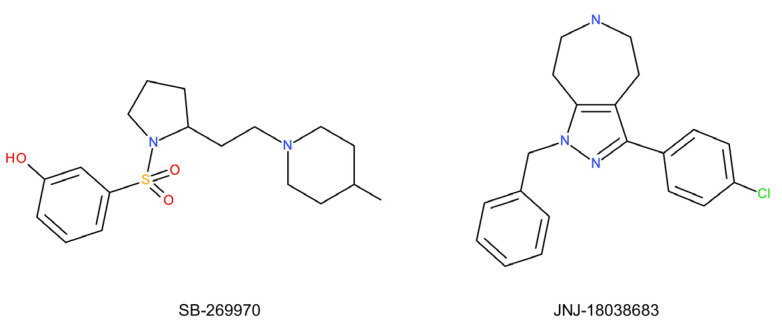
5-HT_7_ receptor antagonists: SB-269970 and JNJ-18038683.

**Table 1 ijms-22-09015-t001:** Clinical effects of augmentation of SERT inhibition with different activities towards 5-HT receptors.

Clinical Intervention	Mechanism of Action	Effect	References
SSRI + pindolol	SERT inhibition + 5-HT_1A_ agonism	Reduced latency of the antidepressant response and improved the clinical response in previously untreated MDD patients	[21]
SSRI + buspirone	SERT inhibition + 5-HT_1A_ partial agonism	Symptom remission in patients unsuccessfully treated with SSRIs	[75]
SSRI + mirtazapine	SERT inhibition + 5-HT_2A_ antagonism	Augmentation of the clinical response to SSRIs in treatment-resistant patients	[76]
Vilazodone	SERT inhibition + 5-HT_1A_ partial agonism	In contrast to prototypical SSRIs, vilazodone has not been associated with treatment-emergent sexual difficulties or dysfunction	[27]
Vortioxetine	SERT, 5-HT_3_ and 5-HT_7_ receptors inhibition, 5-HT_1A_ agonism	Potential rapid onset of action	[77]

## Data Availability

Data are contained within the article.

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
