# Peer review of "5-HT Receptors and the Development of New Antidepressants"

_ijms, 2021, doi:10.3390/ijms22169015_

Round 1

Reviewer 1 Report

Exhaustive paper related to 5-HT receptors and the development of new antidepressants. A lot of work. However I have few suggestions in order to improve the shape of the paper and to facilitate its understanding for the potential readers:

Shape suggestions

As some sections/subsections are very long and hard to follow, maybe the authors will agree to summarise the information provided by tabulating it (having as last column the References) or by making some relevant figures/schemes.

Table 1 is irrelevant, please transform it in text (it provides 1 row of information). Tables are used when many data must be summarised.

4. Conclusions section: please without references. All information already provided in other sections don't need to be reminded. This section just briefly presents the main findings of your own research (after all the information provided in the manuscript were synthesized)

Content suggestions

Abstract. Aim of the study is missing.

1.Introduction. Please highlight as better is possible the aim of the study (as last paragraph of Introduction section). What makes special this study? Which is its novelty character or its special aspects? Why have the authors chosen this topic? What differentiate this paper from others in the same topic? Please make this aim of the study more relevant.

Details were omitted related to the mechanism of depression: please discuss the possible role of FoxO in depression. You may refer to Elucidating the Possible Role of FoxO in Depression. Neurochem. Res. 2021 https://doi.org/10.1007/s11064-021-03364-4 and endocanabinoids role (Integrating endocannabinoids signalling in depression. J Molec Neurosci, 2021. https://doi.org/10.1007/s12031-020-01774-7) ; additionally, regarding the depressive behaviour, I suggest Exploring Sonic Hedgehog Cell Signaling in Neurogenesis: Its Potential Role in Depressive Behavior. Neurochem Res. 2021, ttps://doi.org/10.1007/s11064-021-03307-z and Unfolding the Role of BDNF as a Biomarker for Treatment of Depression. J Molec. Neurosci., 2020 https://doi.org/10.1007/s12031-020-01754-x 

Author Response

Dear Reviewer 1,

Thank you very much for your kind review of the manuscript. All the comments and suggestions are very much appreciated.

The present version of the manuscript was improved according to the suggestions. Detailed comments to each question can be found below.

Specific comments to the Review 1.

Shape suggestions

As some sections/subsections are very long and hard to follow, maybe the authors will agree to summarise the information provided by tabulating it (having as last column the References) or by making some relevant figures/schemes.

Each subsection has been summarised.

Table 1 is irrelevant, please transform it in text (it provides 1 row of information). Tables are used when many data must be summarised.

Table 1 has been transformed in text.

  1. Conclusionssection: please without references. All information already provided in other sections don't need to be reminded. This section just briefly presents the main findings of your own research (after all the information provided in the manuscript were synthesized)

The references have been removed.

Content suggestions

Abstract. Aim of the study is missing.

The aim of the study has been inserted.

1.Introduction. Please highlight as better is possible the aim of the study (as last paragraph of Introduction section). What makes special this study? Which is its novelty character or its special aspects? Why have the authors chosen this topic? What differentiate this paper from others in the same topic? Please make this aim of the study more relevant.

The additional paragraph has been inserted.

Details were omitted related to the mechanism of depression: please discuss the possible role of FoxO in depression. You may refer to Elucidating the Possible Role of FoxO in Depression. Neurochem. Res. 2021 https://doi.org/10.1007/s11064-021-03364-4 and endocanabinoids role (Integrating endocannabinoids signalling in depression. J Molec Neurosci, 2021. https://doi.org/10.1007/s12031-020-01774-7) ; additionally, regarding the depressive behaviour, I suggest Exploring Sonic Hedgehog Cell Signaling in Neurogenesis: Its Potential Role in Depressive Behavior. Neurochem Res. 2021, ttps://doi.org/10.1007/s11064-021-03307-z and Unfolding the Role of BDNF as a Biomarker for Treatment of Depression. J Molec. Neurosci., 2020 https://doi.org/10.1007/s12031-020-01754-x

Citations with a short comment have been inserted.

I do hope that the improved manuscript will fulfill your expectations and requirements for publication. I would like to thank you again for your review, all the comments and suggestions.

Yours faithfully,

Grzegorz Ĺšlifirski

Reviewer 2 Report

This is an interesting review papers.I have several issues to be addressed.

  1. Authors should replace references with recently published papers.
  2. Biological information on 5-HT receptors is not enough. Authors should have one sub-section for giving readers biological information like homolous amino acid sequeces, common 3d structures and domains, etc.
  3. Resolution of chemical structures is not clear. Please replace with clean version.
  4. 338 references are too many. Authors need to delete some old ones and some redundant parts should be deleted to decrease of total pages.
  5. Authors need to prepare Tables summarized with clinically testing drugs and their effects.

Author Response

Dear Reviewer 2,

Thank you very much for your kind review of the manuscript. All the comments and suggestions are very much appreciated.

The present version of the manuscript was improved according to the suggestions. Detailed comments to each question can be found below.

Specific comments to the Review 2.

Authors should replace references with recently published papers.

The references have been replaced.

Biological information on 5-HT receptors is not enough. Authors should have one sub-section for giving readers biological information like homolous amino acid sequeces, common 3d structures and domains, etc.

The article was not primarily focused on the structural analysis of 5-HT receptors. The main insights of the review concern the role of specific 5-HT receptor subtypes due to their distribution in the body and the activity of synthetically obtained molecules in preclinical and clinical studies. Perhaps the analysis of the structure of specific receptor subtypes will be the subject of our future publications, also due to the total number of pages of the reviewed article.

Resolution of chemical structures is not clear. Please replace with clean version.

The figure quality has been improved to 1200DPI.

338 references are too many. Authors need to delete some old ones and some redundant parts should be deleted to decrease of total pages.

The number of references has been reduced.

Authors need to prepare Tables summarized with clinically testing drugs and their effects.

The appropriate table has been inserted. Each subsection has been additionally summarised.

I do hope that the improved manuscript will fulfill your expectations and requirements for publication. I would like to thank you again for your review, all the comments and suggestions.

Yours faithfully,

Grzegorz Ĺšlifirski

Round 2

Reviewer 1 Report

The authors have responded to all my requests.

Reviewer 2 Report

This paper is now acceptable.